# A class for itself? On the worldviews of the new tech elite

Hilke Brockmann[1]*, Wiebke Drews[2], John Torpey[3]

**1** Department of Social Sciences & Humanities, Jacobs University Bremen, Bremen, Germany, **2** Institute for Political Science, Universität der Bundeswehr München, Neubiberg, Germany, **3** The Graduate Center, CUNY, Ralph Bunche Institute for International Studies, New York, NY, United States of America

* h.brockmann@jacobs-university.de

**Data Availability Statement:** All data files are available at https://github.com/wiebkedrews/ClassForItself/releases/tag/v1.0.

**Funding:** The author(s) received no specific funding for this work.

## Abstract

The emergence of a new tech elite in Silicon Valley and beyond raises questions about the economic reach, political influence, and social importance of this group. How do these inordinately influential people think about the world and about our common future? In this paper, we test a) whether members of the tech elite share a common, meritocratic view of the world, b) whether they have a "mission" for the future, and c) how they view democracy as a political system. Our data set consists of information about the 100 richest people in the tech world, according to Forbes, and rests on their published pronouncements on Twitter, as well as on their statements on the websites of their philanthropic endeavors. Automated "bag-of-words" text and sentiment analyses reveal that the tech elite has a more meritocratic view of the world than the general US Twitter-using population. The tech elite also frequently promise to "make the world a better place," but they do not differ from other extremely wealthy people in this respect. However, their relationship to democracy is contradictory. Based on these results, we conclude that the tech elite may be thought of as a "class for itself" in Marx's sense—a social group that shares particular views of the world, which in this case means meritocratic, missionary, and inconsistent democratic ideology.

## Introduction

In the United States, at least, the mid-twentieth century "Great Compression" grounded in a mass-production economy that found good-paying work for much unskilled labor [1] has been replaced by an era of vertiginous gains at the very top of the income distribution and by long-term wage stagnation and declines in wealth for the broad majority of the American population. French economist Thomas Piketty has argued that we are returning to a world similar to the one that preceded the twentieth-century heyday of the manufacturing economy—a world in which returns to capital systematically outpace those to labor [2]. That world has been marked by strong educational demands for competitiveness in the labor market, the "financialization" of the economy, weakened labor unions, lower taxes on the wealthy, toleration of new norms accepting huge disparities in the average pay of executives and ordinary workers, and deregulation of the economy more generally [3].

An important element of this broader economic change has been the much-discussed shift to a high-tech economy. The high-tech economy, at first an almost exclusively American

**Competing interests:** The authors have declared that no competing interests exist.

phenomenon, has gradually come to be dominated by the "Big Nine" (plus one)—Amazon, Apple, Facebook, Google, IBM, and Microsoft in the United States, and Alibaba, Baidu, Huawei, and Tencent in China [4, 5]. Other familiar names in the high-tech firmament include computer hardware manufacturers and software designers such as AmericaOnLine (AOL), Dell, Intel, Oracle, SAP, and Yahoo. More recently, inventors of apps such as Snapchat, Uber, and WhatsApp (now owned by Facebook) have assumed an important place in the global economy. These businesses have remade the economic landscape over the last half-century; none of them other than IBM existed before the late 1960s. The founders and CEOs of these and other tech companies comprise a substantial fraction of the leading members of the tech elite, who have ascended during the last half-century to extraordinary wealth, power, and influence.

In view of their great wealth and social importance, we need to understand more clearly who these people are and how they see the world. This is especially true since scholars have found that politicians are very responsive to the desires of the wealthy people who contribute the vast majority of money for American political campaigns, but typically pay no discernible attention to the concerns of much of the electorate [6]. Who are the new tech elite and what are their social and political views? What is their vision of our common future, and how do they think we will get to that future?

## Background–meritocratic elites, inequalities & views of the future

Research on elites has returned to prominence lately, presumably as a response to the growing economic inequality in the wealthier parts of the world. In many Western countries, a larger share of the social product has been transferred during the past several decades to those at the top of the economic hierarchy than was the case during the period preceding the 1970s. Inequality *within* rich societies has thus strongly increased since the liberalization of markets in the 1980s [see 2, 7, 8], although at the same time income inequality has diminished *across* countries around the globe, largely as a result of the increased share of global income going to China and India [3]. Especially in the aftermath of the financial crisis of 2008, scholarly interest in elite studies thus began a notable resurgence [see e.g., 9, and, for a review, see 10].

Much of the contemporary research on elites is descended from C. Wright Mills' pioneering post-war study, *The Power Elite*. Mills spoke of the elite as comprised of economic, political, and intellectual elites. In a new departure, however, Mills also argued that, as the United States ascended after World War II to global preeminence in the systemic competition with the Soviet Union, the American elite was coming to be dominated by what he called a "military definition of reality" [11]. Echoing Mills' insight, President Dwight D. Eisenhower warned in his 1961 Farewell Address that a new "military-industrial complex" was assuming disproportionate influence in the country—a trend he warned against, despite his own military background and prowess (he had been Supreme Commander of Allied forces in Europe during the Second World War). Despite his concern about the creeping influence of military power, Eisenhower's concern about the challenge from Soviet communism led him to stimulate, through both educational policy and the promotion of research, a series of developments that would help create a new elite in the United States that would gradually come to supplant the old manufacturing elite that dominated the country through approximately the 1970s—the "tech elite."

Notwithstanding the fact that this new elite had originally been extensively "sponsored" [see 12] by the government, the tech elite as we define it here is and understands itself as a strongly "meritocratic" subset of the broader (political, military, economic, ideological/religious) elite. It constitutes a significant and growing proportion (at present approximately one-

fifth) of the world's 100 richest people. As we will show, its members comprise a novel group with distinctive characteristics as compared with other segments of the elite.

First, the tech elite are the private beneficiaries of technologies rich in intellectual property that have been developed over decades in the United States with the aid of considerable public funding [13, 14]. Having been commercialized primarily after their original germination in military- and government-funded research labs, the technologies in question have gradually diffused throughout all areas of social life.

Second, these elites tend to be the products of top-ranked academic institutions such as Stanford, MIT, or Harvard University (U.S. News and World Report Rankings), and are geographically clustered in tech hubs such as Silicon Valley or Shenzhen, China. Economists have demonstrated the positive externalities of geographic proximity, otherwise known as "network effects," particularly for knowledge workers [15–18]. Yet tightly bonded networks may have the vice of their virtues; in addition to the advantages they offer in terms of creativity and mutual recognition, they often also breed incestuous environments and worldviews that may be culturally limited [see 4]. The idea of a social order that allocates rewards based on "merit" has had a remarkable and global career. The term meritocracy was first popularized by the Labour-aligned British sociologist Michael Young in his satire, *The Rise of the Meritocracy, 1870–2033* [19], first published in 1958. Young foresaw a world in which inequality would be legitimized by the combination of "IQ + effort." The result, he feared, would be growing social inequality and, in due course, unrest on the part of the working-class people who would be left out by this scheme. Over time, however, the idea of meritocracy took on a variety of guises [see 20], and the understanding of meritocracy as a just way of distributing social positions tended to win out over the critical view propounded by Young. Over time, neoliberal, market-friendly ideas in politics and the social sciences came to favor equality of opportunity to equality of outcomes. Meritocratic stratification, first understood as just an updated justification of social inequality, thus came to appear just and desirable. Therefore, social scientists began to focus on the problem of how merit-based our societies actually are [e.g., 21–23] and in times of rising inequality lamented declining social mobility [24, 25]. More recently, fueled by the political mobilization of populist parties against the establishment in the US and Europe, the problematic nature of meritocracy has returned to the scholarly and public agenda [26–28]. However, this re-interpretation speaks against the core interests of the tech elite, which may explain why they stick to a positive interpretation of meritocracy.

Third, many believe that rapid technological progress has left democratic checks and balances behind while putting extraordinary wealth and know-how in the hands of the few. Indeed, since the last US election, the public has become increasingly aware of how fake news, hacking, and data manipulation can threaten democratic decision-making [see, e.g., 29–31]. Since 2018, regulators have taken actions in Europe and elsewhere to rein in the activities of the tech giants, which are increasingly seen in the United States as wielders of largely unregulated monopoly power [see 32]. In the aftermath of a horrific attack on two synagogues in New Zealand, demands have grown for tighter regulation of the freedom of speech that is thought to characterize social media platforms [33]. More broadly, the tech giants are now widely said to operate a novel "surveillance capitalism" [34], scooping up enormous amounts of user data and using it to demonstrate to advertisers how effectively they achieve "user engagement" or selling it to third parties at a profit—but destroying "privacy" in the process. Whereas the tech companies had once enjoyed a sort of halo for the whizz-bang products they have created, public perceptions of their activities have grown increasingly cautious; a recent Pew Research Center poll [35] found that "around half of U.S. adults (51%) believe these companies should be regulated more than they are now."

While the social implications of the new technologies may be poorly understood as yet, it is clear that the tech elite have become enormously wealthy and influential in the course of these developments. We hypothesize that these newly wealthy elites are distinctive in comparison with the general public and with other and previous elites. In particular, for an elite their stated views tend unexpectedly toward the egalitarian and "progressive." They frequently profess their desire to "make the world a better place," in a way that is not typically associated with elites in, say, energy production or real estate. Moreover, the philanthropic endeavors of the tech elite suggest that this desire "to make the world a better place" is genuine [36]. We know from such examples as the fraudulent escapades of the Trump charitable foundation [37] that wealthy elites need not be overly concerned with "making the world a better place," so it hardly goes without saying that they do so. But we also know from the scandal concerning fraudulent admissions schemes for getting into top American universities [38] that elites may think they are "making the world a better place" while simultaneously "dream hoarding" [39] and thus entrenching their position and that of their families in the upper reaches of society. Paradoxically, this sort of scheming may be a perverse result of the degree to which today's upper middle classes are dependent on education and performance rather than on inherited wealth for their well-being. To the extent that these "new class" members are not owners of profit-generating capital but have livelihoods that are rooted in their skills and achievements, their situation is more precarious than that of the "old" capitalist class [see 40]. Meanwhile, Mizruchi found that the mid-twentieth century American corporate elite was more supportive of the well-being of the country as a whole than its "fractured" successors in the post-1973 period [41]. This may in part reflect their origins in less securely established and more "meritocratic" strata.

One of the chief aims of this study is thus to adjudicate between two perspectives on the question of the tech elite's commitment to "making the world a better place." On the basis of ethnographic and other evidence, Shapin argues that, notwithstanding the very real concern among tech elites with money as a measuring stick of success, those in the elite world of modern "entrepreneurial technoscience" are sincere about wanting to "make the world a better place" and act frequently on that claim [42]. In contrast and more dramatically, previous research has also portrayed the tech elite as utopian individualists with roots in the counterculture and the *Whole Earth Catalog* [43, 44]; as a prominent segment of the "liberal rich" [36]; and, splitting the difference between these other two perspectives, as "Liberaltarians" whose views are socially liberal (like Democrats) but ill-disposed toward government regulation and unions (like Republicans) [45, 46].

These varied perspectives capture part of the truth about the worldviews of the tech elite. We argue, however, that these individuals are better understood as a new kind of contradictory class stratum. We hypothesize that they are *meritocratic*; *mission-driven*; and *self-misunderstanding* with regard to their place in a democratic order; that is, while they are often genuinely committed to ideals such as democracy and equality, their wealth allows them (and the political system in which they operate in the United States, at least, encourages them) to circumvent democratic processes and to exert disproportionate influence as a result of their extraordinary wealth. Ghiridharadas sees them merely as *bien-pensant* hypocrites who are interested in "making the world a better place" only insofar as it does not undermine their own prerogatives and those of the wealthy more generally [47]. This is too crude, we think, and misses the degree to which these people believe in the goodness of their political and philanthropic activities. We find, by contrast, not hypocrisy but real, often missionary devotion to the many political and philanthropic causes they support. At the same time, members of the tech elite frequently fail to understand how their activities unavoidably (if inadvertently) undermine democratic equality. One might well regard this lack of understanding as a kind of false consciousness—the

false consciousness not of an exploited class, but of a privileged yet in certain respects precarious elite.

Against the background of this literature, we test the following hypotheses:

H1: The members of the tech elite share a common, distinctive, meritocratic view of the world.

H2: Members of the tech elite regard themselves as having a specific mission—"to make the world a better place"—but this may vary on the basis of religious or cultural background.

H3: The members of the tech elite have a contradictory relationship to democracy: they support democratic practices in general but undermine democracy by virtue of the political activities available to them on the basis of their tremendous wealth and influence.

## Data

Elite studies typically face serious methodological challenges. They suffer from their focus on small, highly selected, and difficult-to-access samples. Drawing random samples is inappropriate for analysis of a highly skewed distribution. Accordingly, and in keeping with common practice in elite research [48–50], we take as our research subjects the members of the Forbes list of the 100 richest people in the tech industry [51]. Forbes calculated membership in this top 100 list on the basis of stock prices and exchange rates at the close of business on 18 August 2017. The 100 richest individuals in the tech world were often founders and/or CEOs of the largest companies listed in NASDAQ. The list includes people active in computer hardware, software, social media, venture capital, online gambling, and high-tech manufacturing. The list excludes heirs such as Laureen Powell Jobs and ex-wives such as MacKenzie Scott (formerly Bezos), as well as people in the telecom or media industries. In order to acquire data on the worldviews of the tech elite, we track their statements through the internet. Drawing from Twitter, as well as from the websites of philanthropic foundations, we utilize their own tools to understand their outlooks. Yet reading digital traces is a difficult undertaking. Big data is not designed for research purposes [52], as are interview protocols or survey instruments. It is highly selective. Still, as we have identified our study population, we can address this selectivity. Moreover, our analysis is based on various communication channels, cross-validates results, and tests for the reliability of the analysis.

### Twitter

Founded in 2006, Twitter has established itself as the most popular microblogging service on the Web, where 320 million users post and interact with brief messages known as "tweets" [53]. These may include many of the site's now-familiar technical features, including the famous hashtag (#), retweets (RT) and "at reply" (RE). Twitter is a publicly traded company with strong backing from venture capitalists that, like many of the leading tech businesses, has yet to turn a profit. The pressure to increase revenues while striving to maintain a balance between its competing interests [54] has resulted in numerous changes in functionality in the recent past, including a doubling of the maximum length of tweets from 140 to 280 characters in 2017. Starting with its soft launch at an industry party in Silicon Valley, Twitter's target audience consists particularly of members of the media and tech industries [55]. Twitter is particularly popular among those under 50 and among the college-educated [56]. While users of the platform vary widely in the nature of their engagement, a growing majority of users gets its news from Twitter [57]. In fact, a quarter of Twitter's verified users are journalists, which points to the strong professionalization on the platform [58]. This is not surprising, as Twitter's creators have implemented algorithms on their platform that serve as content filters prioritizing advertising, breaking news, and other popular trends [59]. By favoring some kinds of content over others,

however, the company is acting more and more like a gatekeeper. Twitter is also increasingly intervening in the activities of user accounts and content providers, deleting those it believes to be illegitimate or fake and, thereby, exercising editorial judgment and censorship.

Among those on the Forbes list of the 100 richest people in tech, we identified 30 members with an officially verified English Twitter account. Twitter is blocked in China. However, individuals and companies in China circumvent state-imposed controls and use Twitter through a VPN ("virtual private network," a way of accessing blocked internet platforms and being difficult to identify at the same time).

We scraped Tweets at two points in time (t1 = Nov 30, 2018, and t2 = March 8, 2019) and collected back in time the maximum number of unique Tweets per account, which resulted in 49,790 Tweets of all 30 account owners from the tech elite. We also collected the same number of Tweets from a random sample of the general US Twitter-using population. We used the R package streamR [60].

### Individual philanthropic foundations and the giving pledge

Many of the billionaire members of the tech elite are engaged in philanthropic endeavors, often through their own foundations. Charitable activities implement the values and worldviews of their usually wealthy founders. In 2017, US foundations increased their spending by 6.0% (3.8% in real terms) over the previous year. These foundations contributed 16% of all charitable giving in the US, for a total of $65.6 billion, of which 31% went to religious institutions, 14% to education, 12% to human services, 11% to other foundations, 9% to health, 7% to public-society benefit, 6% to international affairs, 5% to arts, culture, and the humanities, 3% to the environment and animals, and 2% to individuals [61].

We identify 60 private foundations set up by members of the wealthiest 100 people in the tech world. In the United States, private foundations are special legal entities (501c3 organizations, according to the tax code) that pursue a good (public) cause and, in consideration of those aims, receive tax exemptions. We scraped the mission statements of these websites as one source of information on the worldviews and future visions of the tech elite.

The Giving Pledge (https://givingpledge.org/) is a philanthropic initiative of Warren Buffett and of Bill and Melinda Gates. Signatories pledge to give at least half of their net wealth to charity or to other philanthropic projects before they die. We identify 17 tech billionaires from our list (out of a total of 211 signatories) who have made the commitment by signing the Giving Pledge. The texts of their commitment statements comprise another source of data for our analysis.

### Socio-demographic sources

Furthermore, we collected and added basic socio-demographic information (age, sex, nationality, place of residence, education, marital status, number of children) from the Forbes 2017, Hurun Rich List 2017 [62], and Inside Philanthropy websites [63] (accessed April 2018). The latter is a private initiative created by the analyst David Callahan [see 64] aiming to bring greater transparency to the world of philanthropy. Inside Philanthropy maintains an extensive database of private philanthropic foundations. In case of missing information, which was marginal, we relied on Wikipedia. This socio-demographic data was triangulated to the social media data.

## Methods

### Automated text-analysis

Our analysis is based on automated text-analysis, sentiment analysis and machine-learning approaches. We use a common "bag-of-words" model (BOW) which focuses on the frequency

of the occurrence of words but ignores context and grammar. Until recently, BOW has been applied in nearly all automated text and sentiment analyses [e.g., 65–67]. New studies have discovered the richer linguistic structure beyond BOW [e.g., 68, 69], but "achieving improvements over these simple baselines can be quite difficult" [70]. We decided in favor of the baseline approach and blank out the embeddedness of words for three reasons: Firstly, due to the scarcity and selectivity of social media data on the tech elite, we use multiple data sources to back up our findings. Because online communication varies by platform, we needed a simple, comparable approach. Moreover and secondly, there is no theoretically informed approach concerning (tech) elite online communication that provides advice on how to expand a baseline BOW model. Pioneering into the field of tech elite online conversation, we also make use of approaches that do not focus on digital communication to test our hypotheses on the tech elite as a distinct social class.

For our analyses we use R, and especially the package "quanteda" [71], which is commonly used in text analysis due to its fast and easy manipulation of texts in a given corpus by performing the most common natural language processing tasks. We preprocessed the data, excluded stopwords, created a document-feature matrix, and calculated descriptive, inferential statistics for certain words and sentiments and supervised machine learning models. To examine the distinctiveness of our study population, we applied a dictionary approach to measure the prevalence and use of words relating to "merit" and "democracy". We used Merriam-Webster (https://www.merriam-webster.com/thesaurus) to index and gather *all* synonyms, related words, and idiomatic phrases as they are used in everyday language. We do not refer to theoretical definitions, to normative assumptions or other specifications that each concept may entail. Thereby, we can match patterns of some words listed by Merriam-Webster with so-called "glob"-style wildcard expressions that pick up on numerous words with identical word stems.

The *merit dictionary* is comprised of "merit*", "cardinal virtue", "distinct*", "excellen*", "grace", "value", "virtue", "advantage", "edge", "plus", "superiority", "account", "valuation", "worth", "assessment", "estimation", "evaluation", "great*", "perfect*", "consequence", "importan*", "significan*", "weight", "desirability", "deserve", "earn", "rate", "entitle", "qualify".

The *democracy dictionary*, in turn, includes "democ*", "demos", "republic", "self-government", "self-rule", "pure democracy", "home rule", "self-determination", "autonomy", "sovereignty", "sovranty", "popular", "republican", "self-governing", "self-ruling", "represent*", "libertarian", "nontotalitarian", "demonstrations", "rallies", "assembl*", "conferenc*", "congress*", "convention*", "convocation*", "council*", "gathering*", "march*", "protest*", "sit-down*", "sit-in*", "strike*", "counterdemonstration*", "counter-demonstration*", "counter-protest*", "counter-protest*", "counterrall*", "counter-rall*".

For all other concepts, we have used the pre-defined LIWC2015 dictionary (http://liwc.wpengine.com/). The LIWC2015 is a comprehensive dictionary developed by psychologist James Pennebaker. It is a tool widely applied in automated text analysis. It includes an "achieve" category which we use to cross-validate the findings for "merit", which, like "democracy", was not featured. The LIWC2015 also has a category for religion and a focus on power and money. These categories allow us to test hypotheses 2 ("make the world a better place") and 3 (tech elites' relationship to democracy). Most importantly, the LIWC2015 entails a long list of positive or negative emotions. We count the emotional valence of the text and of subtexts and compare them with those of either the random US population (tweets), other super-rich people (signatories of the Giving Pledge), or different age cohorts (foundations) to test whether the tech elite communicates differently than these other populations.

Emotions are significant components of speech, as they are "short-lived experiences that produce coordinated changes in people's thoughts, actions, and physiological responses" [72].

When people communicate emotionally, they attach importance to certain subjects and intend to steer attention, influence decision making, and guide the behavior of their audience. Sentiment analysis is a text mining method that has been applied widely in marketing to determine individuals' perceptions of a product, and in computational social science to learn about social, economic, and political issues and dynamics, often in real-time. Sentiment analysis performed on data gathered from social media platforms constitutes a methodological alternative to classical survey research. Its efficiency has been demonstrated especially in the study of electoral campaigns [e.g., 66, 73] and in researching political preferences [74]. The approach is less applied to elite studies, particularly outside the political sphere. The novelty of this study is to understand the sentimental loading of the above-mentioned concepts and to test for significant differences within and between the tech elite, other elite donors and the general population.

## Machine learning

To determine if social media communication is fit and robust for elite research, we examine if we can distinguish the tech elite from the Twitter population. First and as a baseline, we run a logistic regression and regress class membership (tech elite vs US population) on the aforementioned content categories (future, merit, achieve, religion, democracy, power, money) as well as on positive and negative sentiments. Then we apply supervised machine learning techniques to classify Tweets in an automated process. Naïve Bayesian classifier (NB), Support Vector Machines (SVM), and Random Forests (RF) are commonly applied machine learning methods for text classification [see, e.g., 75]. We use NB, SVMs with linear kernel and with polynomial kernel, and fast implementation of RF for high dimensional data (R package ranger).

NB is a probabilistic classifier, for a document $d$, i.e., a Tweet, out of all classes $c \in C$. The classifier returns the estimate of the correct class $č$ which has the maximum posterior probability given the document. In other words, it calculates the most probable class $č$ given some document $d$ by choosing the class which has the highest product of two probabilities—the prior probability of the class $P(c)$ and the likelihood of the document $P(d|c)$—divided by the probability of the data (irrespective of the class) $P(d)$ [76]. Thus, NB treats features, such as the words in our Tweets, independently, and is making a simplifying, "naïve" assumption for real-world examples: $č = (P(d|c) P(c)) / P(d)$.

SVM, in contrast, is a geometric model classifying texts by looking at their position on a hyperplane that separates and classifies a set of documents, i.e., Tweets, into two (or more) classes. Support vectors are the data points closest to the hyperplane, and the most similar examples between classes. By learning the similarities between documents, SVM maximizes the line with the widest margin that separates the respective classes [77]. SVM relies on kernel classifiers which, for text classification, commonly is a linear algorithm because texts are often linearly separable [78], and linear kernels are easier and faster to compute [79]. We also test polynomial kernels.

RF is an ensemble-based classifier that focuses on multiple decision trees to construct powerful prediction models. The method links bootstrap aggregation (bagging) with random feature selection to improve and decorrelate decision trees. After a random forest is generated, trees' predictions are combined to classify new objects [80].

RF performs well on many problems, and can handle noisy and large data sets. However, they are not easy to interpret, which is why we use them here only as an additional validation measure. In comparison, NB performs particularly well with fewer training cases and short documents. SVM, provides better classification tasks with a large amount of training cases

[81]. Our dataset consists of short but many texts. For these reasons, we run and compare different classifiers.

Each classifier is trained on class labels (Tech elite, US population) with a randomly partitioned subset of Twitter data (training set, 80%), and predicts the most likely classes with the remaining unlabeled Tweets (test set, 20%). Moreover, to ensure the robustness of the SMV classifiers, each classification process has been repeated 10 times using different random splits into training and test data. A grid of parameter values was evaluated using cross-validation to obtain the optimally tuned SVM model. Finally, all classification results are evaluated by standard performance measures (accuracy, precision, recall), visualizations (ROC curve), and statistical tests (ANOVA). The code is published on Github (https://github.com/wiebkedrews/ClassForItself/releases/tag/v1.0).

## Results

### Descriptives: Wealth and clusters

The wealth of the tech elite is truly staggering. In 2017, the 100 richest people in tech had a net worth of US$1.081 trillion, or 0.4% of the total global wealth counted in that year. Their average wealth is nearly 25,000 times higher than that of an average American or Canadian, and nearly 69,000 times higher than that of an average European adult [82].

Those on the list made their money primarily in computer software, hardware, and internet-related technologies and services. Founders, executives, and investors in such companies as Facebook, Google, Amazon, Airbnb, eBay, and Microsoft are prominently represented. Such persons comprise 19 out of the total of the 100 richest people in tech.

The tech elite consists mainly of middle-aged men from an economic superpower. Of the top 100, fully 94 are men and only 6 are women. Their average age is 54 years. Half are Americans by nationality; 17 come from China, three from Hong Kong, and a total of seven more come from other parts of East Asia: South Korea, Japan, Taiwan, and Singapore. Like Canada, Europe is home to 5 of the top 100 tech billionaires. Three come from Israel, two each from India and Australia, and one each from Brazil and Russia. Data on citizenship was lacking for one case.

A world map highlights spatial clusters of residence. The tech elite lives overwhelmingly in the global north (Fig 1), and primarily on the East and West Coasts of the United States as well as on China's East Coast.

The superstars of tech also share similar educational backgrounds. Elite American institutions of higher education play a decisive role: Harvard (13) and Stanford (10) were the most frequently attended universities. Yet 22 of the top 100 never studied at a college or university; one case is missing. The most common degree is an undergraduate BA or BSc (35). Thus, the completion rate (78%) is noticeably higher than for the general American population (60% for a bachelor's degree in 2016) [83], despite the inclusion in this group of some famous drop-outs, such as Bill Gates (12). For the majority who attended a college or a university, the most common majors were engineering (28), business administration (13), and computer science (9). Altogether, and somewhat surprisingly, only 46 of the top 100 have an educational background in STEM fields. Finally, twenty of the top 100 earned a master's degree, 14 completed an MBA or EMBA, and 10 had doctoral degrees. Moreover, more than three quarters of the people in our sample (77) are married or remarried, 9 are single, 8 divorced, and one lives in a partnership. For 23 persons, we do not have information on children. The rest have on average two children.

### Visions in the web

**Twitter.** To understand whether and to what extent the worldviews of the tech elite differ from those of other people, we use as a first benchmark a random sample of the American

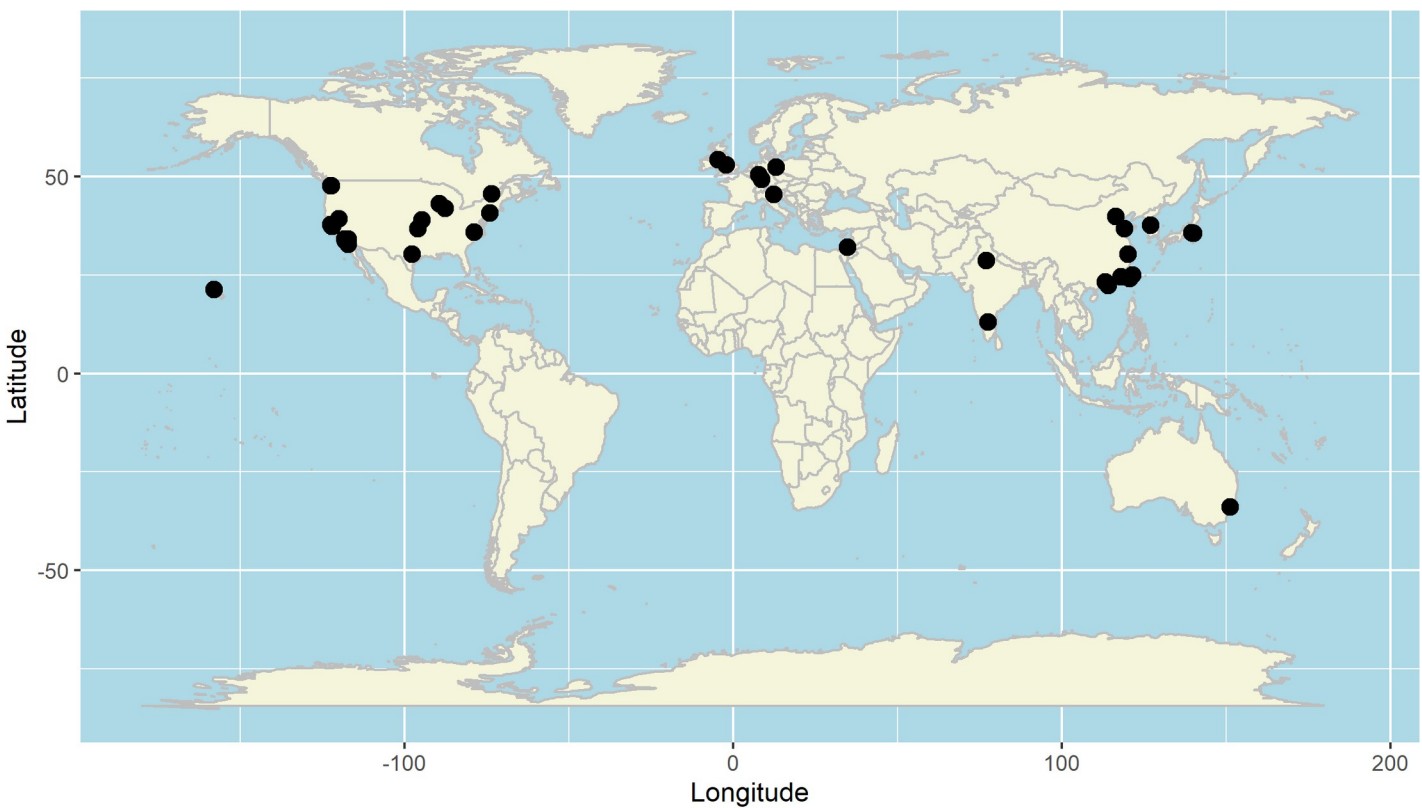

**Fig 1. Global clusters of tech wealth.**

Twitter-using population. We begin with an analysis of simple word frequencies from each of the Twitter samples. The word cloud generated from these word frequencies (see Fig 2) illustrates the fact that the general population of tweeting Americans engages in a more self-referential, judgmental conversation concerning individual, short-term activities than does the tech elite. The words most frequently used by the general population are "like", "get", and "just", followed by "job", "go", "see", "one", "day", "can", "time", "love", "work", "know", "look", and "good". Evaluations are mostly positive ("good", "great", "right"), and more rarely negative ("never").

In contrast to this breezy, mundane pattern of chatter, disruption is consistently at the core of the communications of the tech elite (see Fig 3). As compared to the general tweeting population of the US, they also use more positive words to convey their messages and, as one might expect, they refer much more frequently to peers and to tech firms. The most frequently used words are "new" and "great." More important than "like" or "work" is the word "can", which reflects more optimism about future potential and beliefs than actual activity. Our tech elites also frequently use temporal words such as "today", "day", "year", "years", and "future", and they refer more often to a larger audience such as "people", "us", "team" or "world". Their judgmental language tilts toward the positive, with words such as "good", "better", "best", and "right", and tends to lack negative adjectives. Firm brands and tech leaders are also prominently represented by such terms as "airbnb", "uber", "salesforce", "dell", and "tesla".

Future topics raised more frequently by the tech elite than by the general tweeting population of the US are distinctively connected with tech firms and entrepreneurs (Fig 4). Ordinary US Twitter users relate future words to the current "season," to "Jesus," or to the current US

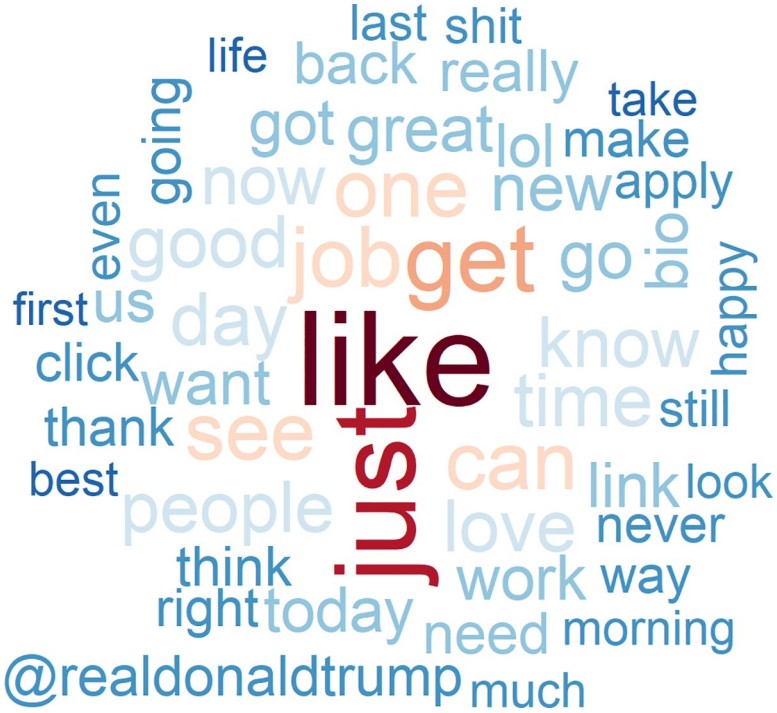

**Fig 2. The 50 most-often used words in tweets of the general US Twitter-using population (N = 49,790).**

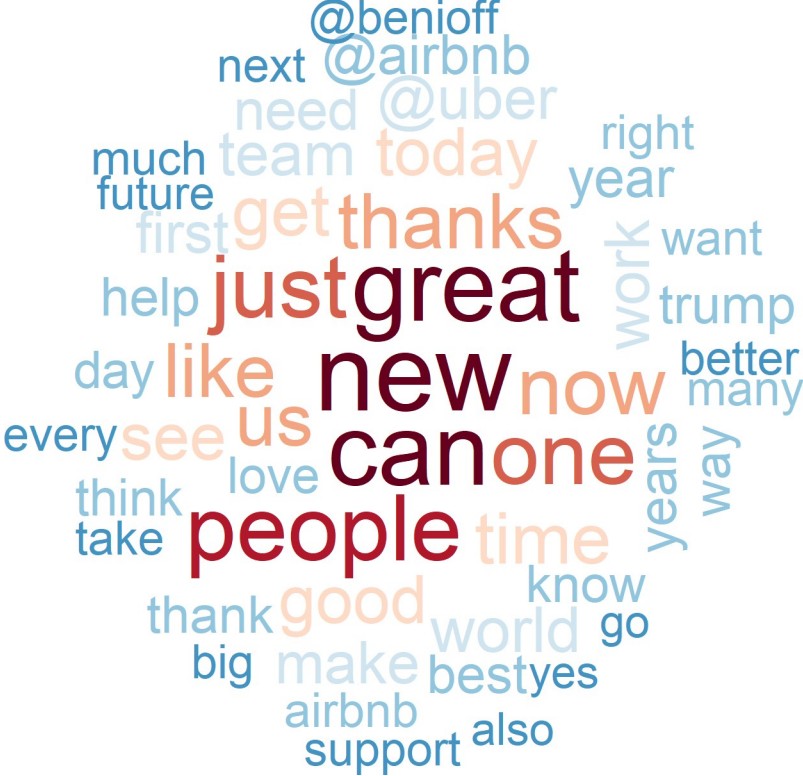

**Fig 3. The 50 most often used words in tweets of the tech elite (N = 49,790).**

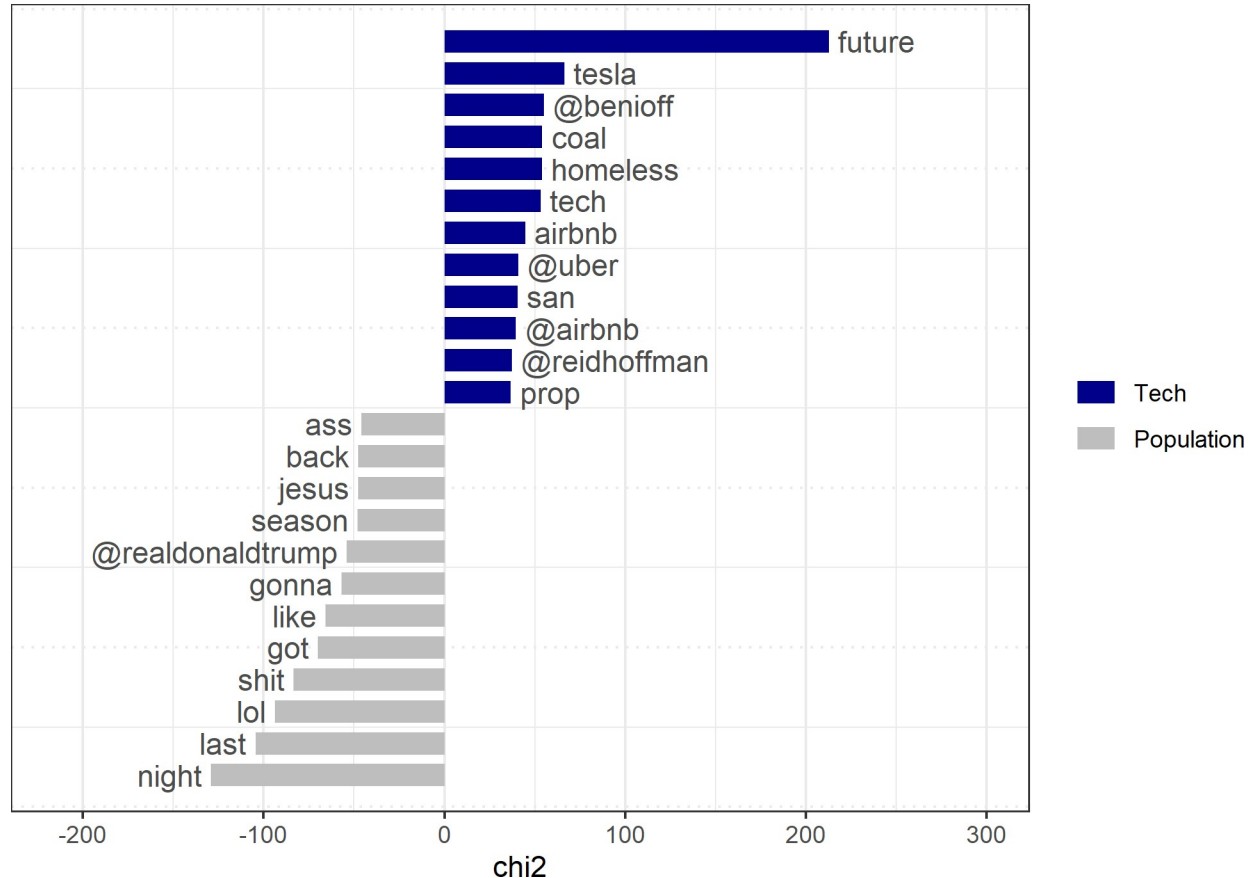

**Fig 4. Keyness statistics for tweets with future-related words.**

president ("@realdonaldtrump") and to personal and immediate experiences (e.g., "back," "like," "got," "lol," "last," "night"). Moreover, when we filter Tweets that contain future-related words, we see that the population systematically frames the future in more negative and colloquial terms ("ass," "shit") than do the members of the tech elite ($t$ = -8.88, $p$ < 0.000).

The tech elite's distinctive attitudes toward the future should also be reflected in a different mode of communication concerning social values, particularly values that legitimate social stratification and elitism. A dictionary which contains the term "merit" and its synonyms shows that members of the tech elite are more than twice as likely as the general population to invoke merit-related words (2,267 versus 4,633 tweets). Testing for the most distinct keywords, we see again in the conversation on merit that business-oriented argot is considerably more prevalent in the tweets of members of the tech elite than it is in those of the general US Twitter-using population. The latter tend to connect merit to employment-related words ("referrals", "hr", "apply", "bio", "job") (Fig 5).

Moreover, the two divergent modes of conversation on merit are differently anchored in emotional terms. The 100 richest people in tech tend to choose both significantly more positive sentiments ($t$ = 3.09, $p$ < 0.010) and less negative sentiments ($t$ = -8.56; $p$ < 0.00) than the general US Twitter-using population.

To cross-validate the dictionary and the findings, we repeat our analysis with the widely used LIWC dictionary and its "achieve" category. The list of synonyms in the LIWC is longer than for the "merit" category (213 words vs 24 words and 5 regular expressions) and the

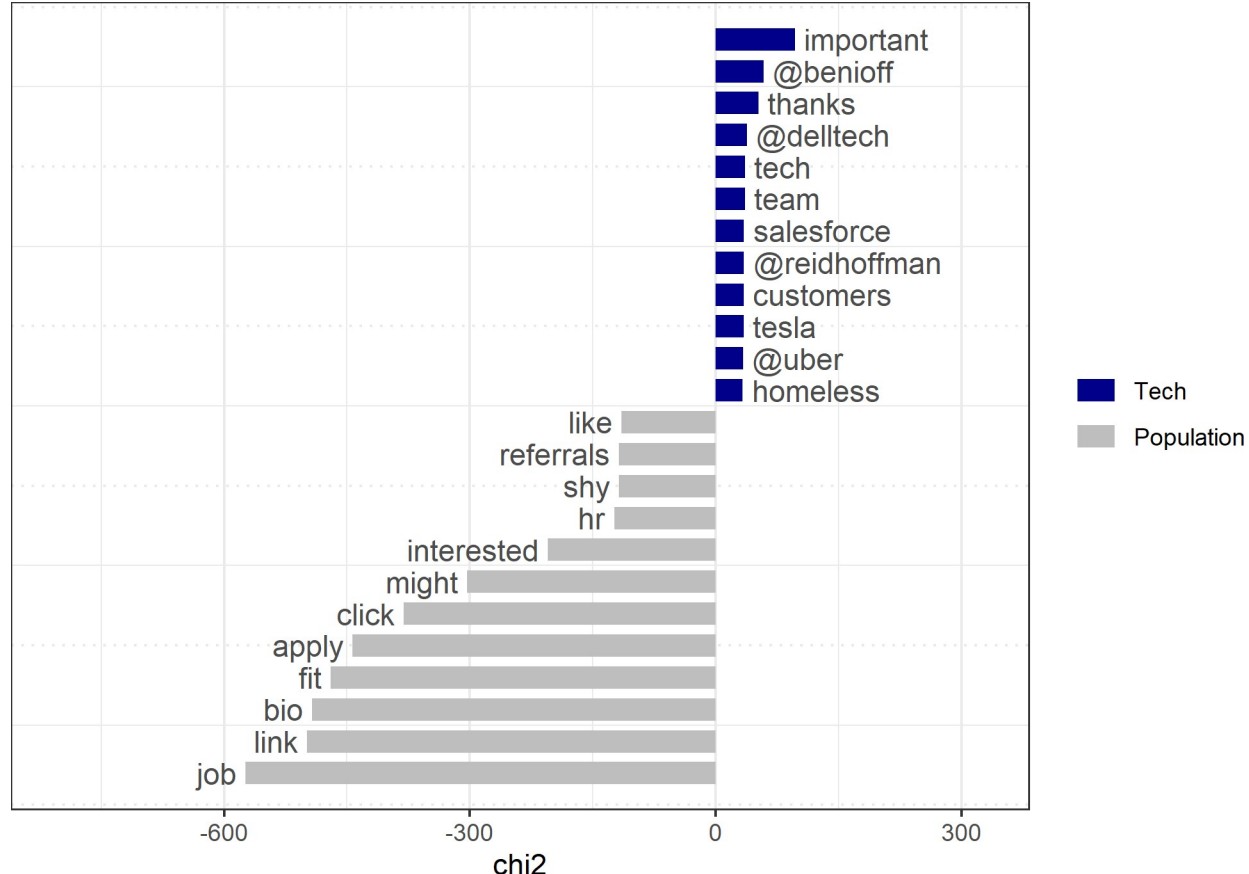

**Fig 5. Keyness statistics for tweets with "merit" and related words.**

former also contains antonyms. Despite these different measures, we see similar patterns. Members of the tech elite use words that fall into the "achieve" category almost twice as often as the general population. In total, the tech elite used 19,431 achievement-related words in their tweets, whereas the random US Twitter-using population mentioned achievement words only 9,439 times ($t = 50.43$, $p < 0.000$).

The ranking of the most distinct key features in the two groups also reproduces well-known differences between them (Fig 6), such as the previously noted references by the tech elite to firms and tech firm founders, on the one hand, and the references to personal job-related experiences on the part of the general population, on the other.

A sentiment analysis reveals an even stronger gap in the diction of the two groups. Members of the tech elite choose more positive words when they talk about achieving, achievements, and related topics than do random US Twitter users ($t = 7.80$, $p < 0.000$). They also refrain significantly more often from indicating negative emotions ($t = -13.68$, $p < 0.000$). Our first hypothesis is thus confirmed.

**The giving pledge and charitable foundations.** Our second hypothesis holds that highly successful tech entrepreneurs have a specific mission and vision for the future. In order to assess this hypothesis, we compare the mission statements associated with the philanthropic vehicles of members of the tech elite with those of other very wealthy people, and we compare those of different tech cohorts to one another.

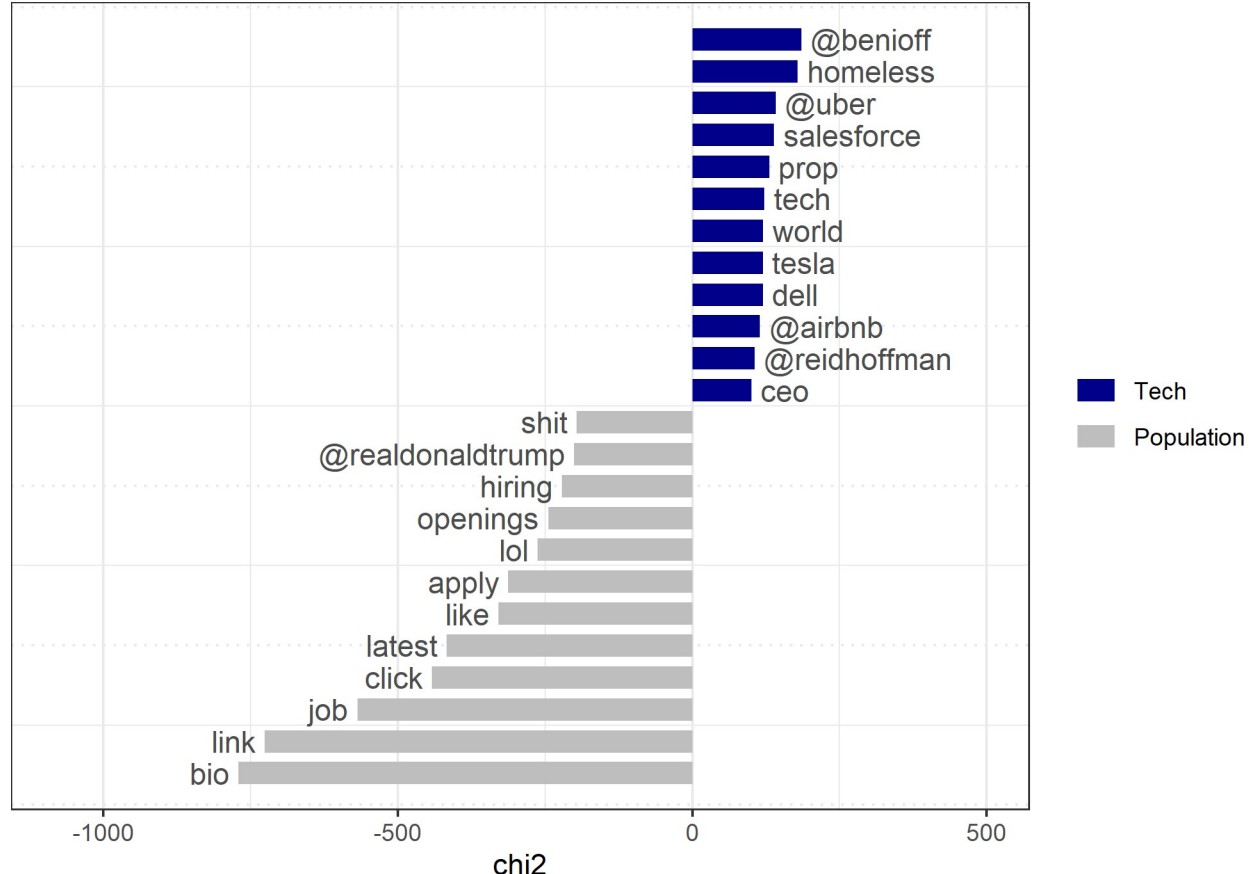

**Fig 6. Keyness statistics for tweets with "achieve" and related words.**

In our sample of donors to the Giving Pledge, we identify 17 members of the tech elite who explain in a "commitment letter" to the project initiators why they have joined the effort. It is perhaps noteworthy that these letters are, on average, shorter for the tech elite (1,796 words) than they are for other wealthy signatories (2,422 words). A word cloud of the 50 words most frequently used by members of the tech elite and by other wealthy donors reveals a more coherent choice of words among the tech elite. Thus, their cloud is larger (Figs 7 and 8). The terms "education", "work" and "social" are at the core of the commitment letters submitted by members of the tech elite. Tech elite members also promote agency in their mission statements, using verbs such as "believe", "create", "put", "grow", "help", and "feel", as well as the verbal "promoting". Moreover, they aim to achieve"progress" and "impact", and invoke the "public", "community", "systems", "children", "healthcare", and "science". Other signatories use their personal experience as a benchmark to evaluate or compare things ("like"), refer less specifically to "many" or to "one" and have "business", "family" and "life" as activity domains on their radar.

The more factual mode of communication that one finds in the statements submitted by members of the tech elite is not necessarily mirrored in a less emotionally-laden writing style. To be sure, compared to other signatories of the Giving Pledge, they add significantly fewer positive emotions ($t = -10.46$, $p < 0.000$) and fewer negative emotions ($t = -6.18$, $p = 0.000$) as they explain why they want to give away at least half of their wealth. If we also take account of the length of their letters, however, quite the opposite is true: members of the tech elite

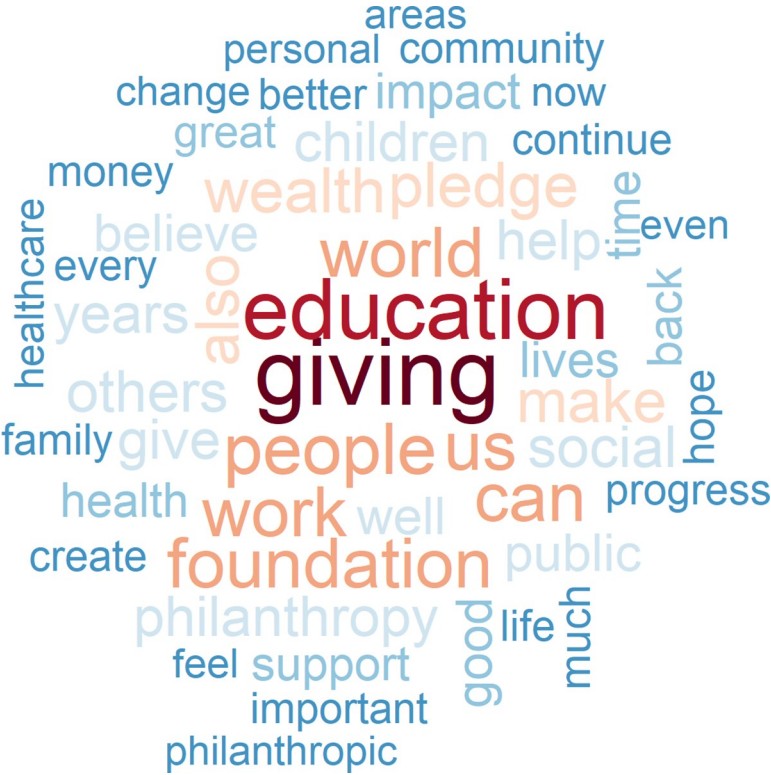

**Fig 7. The 50 most frequently used words in the commitment letters submitted to the giving pledge by the tech elite versus no tech elite.**

communicate even more positive emotions and fewer negative emotions per word than other signatories of the Giving Pledge. The first part of Hypothesis 2 is thus confirmed.

Emotions are drivers for action. Do members of the tech elite have what one might regard as a mission, and perhaps even a "religious" mission? The LIWC dictionary contains religious words as a category. If we apply this category of words to the commitment letters of the signatories of the Giving Pledge, we see that IT entrepreneurs employ only half as many of the religious terms used by other wealthy donors in their pledge statements (proportional word count 0.004 versus 0.002). Religious motives thus play a minor role for the members of the tech elite. On the basis of the data gathered from the commitment letters submitted to the Giving Pledge, we find evidence that the tech elite promotes more secular ideas about how to "make the world a better place for future generations" (Mark Zuckerberg) or to "contribute significantly to try and create a better world for the millions who are far less privileged" (Azim Premji) than other signers of the Pledge do.

We try to cross-validate this finding with the mission statements published by private philanthropic foundations. Sixty tech entrepreneurs from our sample have charitable foundations which maintain their own websites. Mission statements varying between 110 and 4,871 words ($x$ = 684 words) disclose the aims and values of the foundation and, presumably, of its founder (s). We compare different cohorts of the tech elite to test whether they share a coherent mission.

The following word cloud (Fig 9) highlights the 50 most prominent words in these mission statements. The terms "foundation," "education" and "research" lead the list. Related nouns such as "development", "technology", "school", "university", and "program" underline the

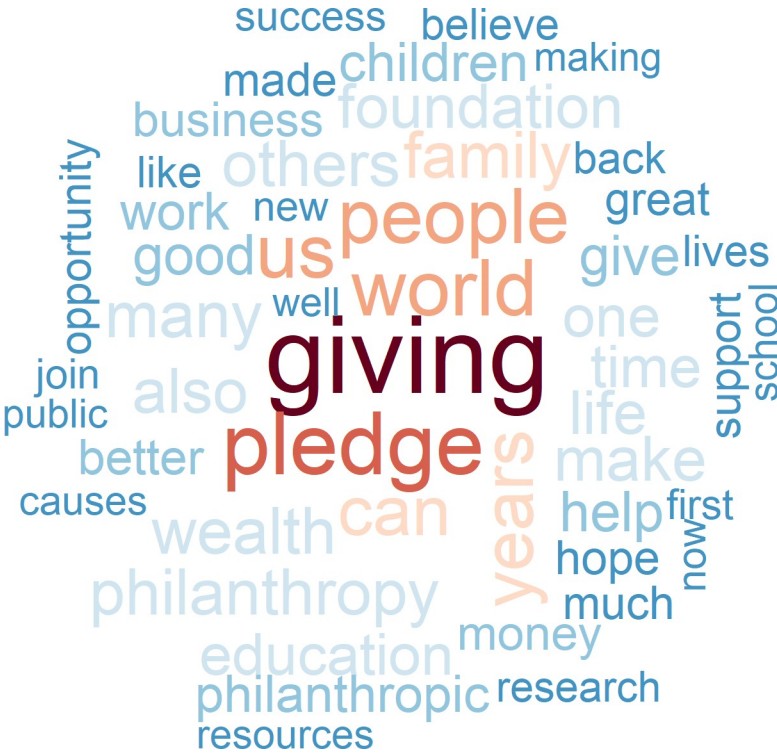

**Fig 8. The 50 most frequently used words in the commitment letters submitted to the giving pledge by the tech elite versus no tech elite.**

importance of educational objectives. Moreover, it is often suggested in the mission statements that these efforts should reach the entire "world", a "global" audience of "people", and "children". Moreover, positive and pro-social activities like "can", "support", "impact", "help", "change", "work", "create", and "provide" signal a positive and feasible mission. This impression is also underlined by positive adjectives like "high", "new", and "social".

For comparison, we distinguish tech cohorts from each other. The evolution of IT technology came in three waves and we identify them by the age of the entrepreneur. The first innovators developed semiconductors and represent the "hardware" cohort (n = 16) (<65 years). The second innovation wave changed the "software" of computing (n = 22) (45–64 years). And finally, the most recent innovation and the youngest cohort is represented by "internet" entrepreneurs (n = 11) (> = 44). By dividing our "bag-of-words" according to these groups, we see how the mission target expands. The most frequently used term in the philanthropic mission statements of "hardware" innovators is "research"; for "software" representatives, it is "school"; and for the "internet" elite, it is "can" (Fig 10).

A sentiment analysis reveals further that the "internet cohort" communicates its mission in the most emotional and positive way compared to the older cohorts of entrepreneurs ($x = 3.7$ versus 2.2 for the "software cohort", 0.7 for the "hardware cohort"). The differences between the internet and the hardware cohort are statistically significant ($p<0.000$), despite the small cohort size. Religious terms hardly appear in mission statements of the tech elite; the "internet cohort" uses none of them at all. All differences between cohorts are insignificant. Insofar as we can infer a religious leaning or background from these mission statements, we find no empirical evidence that missions vary with respect to a certain religious conviction. This falsifies the second part of our hypothesis 2.

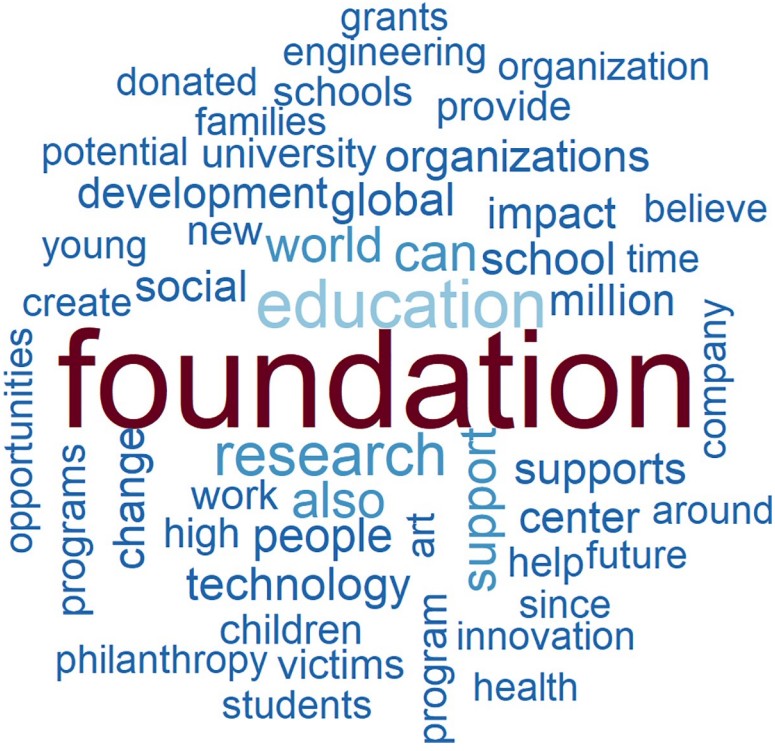

**Fig 9. The 50 most frequently used words in the mission statements of the philanthropic foundations of the tech elite.**

Word clouds also reveal which topics and references have been mentioned too infrequently to appear in the cloud. Notably, there is no mention of the "state" or of "government", even though education and research are among the chief responsibilities of modern governments. There are also no references to democratic procedures, which are typically involved when "support", "change", and "impact" are enacted on a large ("global") scale.

**Views of the tech elite concerning democracy.** In order to understand the tech elite's particular relationship with democracy, we compare their communication behavior on Twitter again with ordinary users. Both groups, the tech elite and the general population, pay comparatively little attention to topics related to democracy. The former mention democracy and related words in 1,359 or 2.7% of their Tweets, the latter in only 710 (1.4%) Tweets. Despite the lack of consideration of these issues, representatives of the tech elite still employ a distinct type of language in regard to democracy. First, they point significantly more often to abstract democratic notions such as "request", "popular", or "democracy". Second, they refer to more abstract political groups like "nazis", or "world". And finally they connect this talk to firms and personalities from their industry, such as "airbnb", "uber", "tesla" or (Marc) "benioff", the founder of Salesforce. In contrast, Twitter users from the US population personalize their messages, refer to the US "Democrats", or to "representatives" of the Republican party such as two-time Attorney General William Barr or the US president. Information is usually not first-hand information but "reports", and "clickable" "links" (Fig 11).

A sentiment analysis further shows that the richest 100 people in the tech industry communicate slightly less positively ($t = -1.69$, $p = 0.090$) and significantly less negatively about democratic issues than the general US Twitter-using population ($t = -3.50$, $p<0.001$). Being less emotional when referring to "democracy" may mean that the elite is less interested in steering

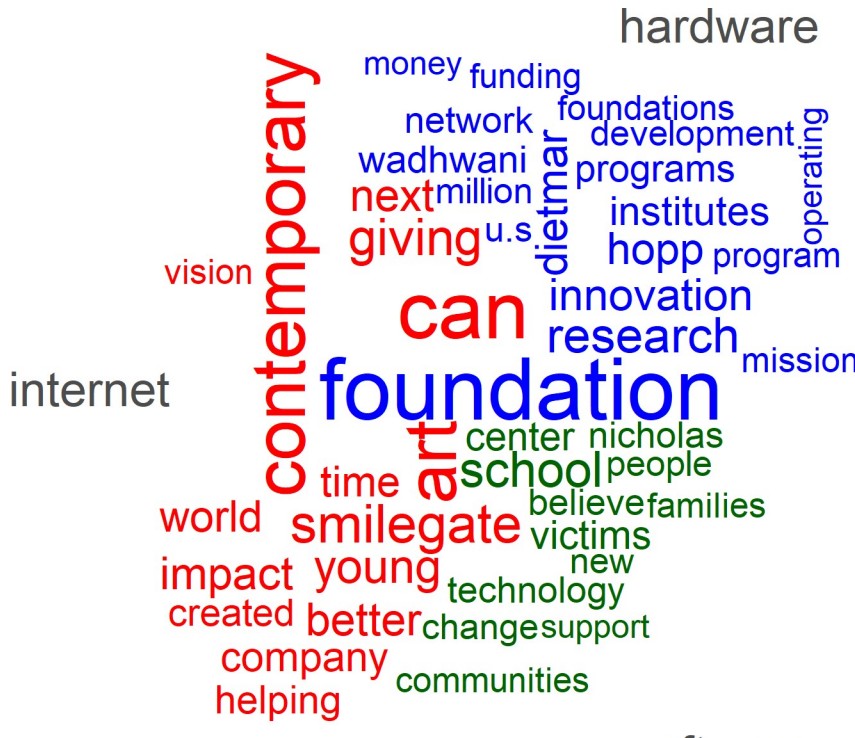

**Fig 10. The 50 most frequently used words in foundations' mission statements by tech cohort.**

attention, influencing decision-making, or guiding the behavior of their audience on that political issue.

To explicitly test the hypothesis that the tech elite has a contradictory relationship with democratic governance, we interrogate the connection between Tweets focusing on democracy, power and money. If we use all Tweets, the mean word scores for power ($t = 47.39$, $p > 0.000$) and money ($t = 37.31$, $p < 0.000$) are significantly higher for the tech super-elite than for ordinary Twitter users.

However, within the selected sample of Tweets that mention democracy, the tech elite refers significantly less often to money (tech group mean of 0.2 money words per democracy Tweet versus 0.3 money words for the Twitter population, $t = -2.76$, $p = 0.006$) and not substantially more often to power (tech elite group mean of 1.1 versus 0.99 power words in the Twitter population, $t = 1.96$, $p = 0.05$). As representatives of an economic elite, they do not see or do not want to communicate a connection between these components of social potency. Moreover, correlations between democracy, money and power further reveal the logically inconsistent communication of the tech elite (Table 1). In fact, while money and power as well as power and democracy correlate positively in the Twitter conversation of the tech elite as well as that of ordinary Twitter users, the former dispute a positive connection between money and democracy that would logically result from the previous positive connections, while the latter see this connection clearly.

This inconsistent communication about democracy, money, and power speaks to the contradictory relationship of the tech elite with democracy, and it does not change if we restrict our sample to US tech entrepreneurs. Our third hypothesis is confirmed.

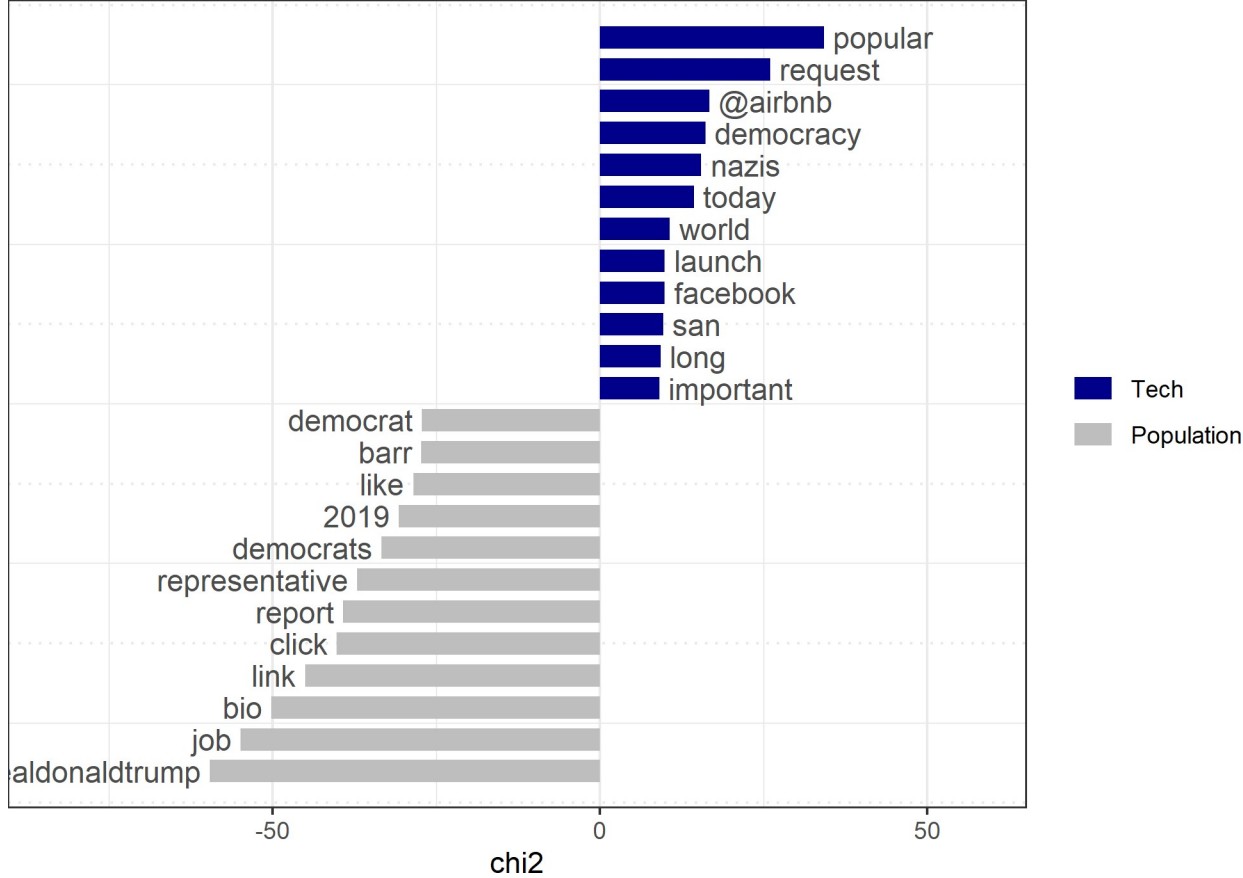

**Fig 11. Keyness statistics for tweets for "democracy" and related words.**

## Classification

In a final step, we test how well we can identify the class of the tech elite (vs non-tech elite). First, a logistic regression model reveals that all of the hypothesized topics (future, merit, achieve, religion, democracy, power, money, positive and negative emotions) are highly statistically significant in the determination of class membership ($p<0.001$). Nonetheless, all topics together predict only 5% of the overall variance ($R^2 = .048$) [S1 Table]. We then test if supervised machine learning approaches with NB, SVM, and RF classifiers can predict if a person

**Table 1. Correlation between democracy, power and money in tweets of the tech elite and the US Twitter population.**

|  | Tech Elite | Population |
|---|---|---|
| **Power & Money** | 0.21 | 0.17 |
|  | *(48.08)* | *(36.68)* |
| **Power & Democracy** | 0.11 | 0.12 |
|  | *(23.66)* | *(26.73)* |
| **Money & Democracy** | 0.01 | 0.05 |
|  | *(-1.05)* | *(11.13)* |

Note: t-values in brackets

**Table 2. Confusion matrix and performance measures for classified Twitter data.**

| | Naïve Bayesian (N = 100.000) | | Linear SVM (N = 50.000) | | Random Forest (N = 100.000) | |
|---|---|---|---|---|---|---|
| | **Tech** | **Population** | **Tech** | **Population** | **Tech** | **Population** |
| Tech | 9237 | 829 | 3710 | 670 | 7944 | 2176 |
| Population | 2391 | 7474 | 1278 | 4342 | 1988 | 7823 |
| Accuracy | 0.8384 | | 0.8052 | | 0.7911 | |
| Sensitivity | 0.7944 | | 0.7726 | | 0.7998 | |
| Specificity | 0.9002 | | 0.8663 | | 0.7824 | |
| AUC | 0.8376 | | 0.9313 | | 0.7911 | |

belongs to the tech elite or to the US population (for more details, see Methods section) without further content. For NB, we used a training set with 80% of the original Tweets. The confusion matrix in Table 2 displays the number of correctly and falsely classified Tweets. The overall accuracy is 83.8%, 91.8% of Tweets are predicted precisely (PPV not shown in the table) with a recall or sensitivity of 79.4%.

In comparison, the results for SVM with a linear kernel reveal similar numbers, even though we use only half of the cases for computational reasons. Overall, accuracy lies at 80.5% with a slightly lower PPV of 84.7% and a recall of 77.3%.

Performance measures for a random forest classifier differ marginally. An ANOVA reveals no significant difference between the three classifiers ($F = 1.24$, $p = 0.35$). Only SVMs with polynomial kernel perform weakly why we excluded them from the further analysis. Our classifiers achieved an average area under the receiver operating characteristic curve (ROC) of 85.3 (AUC) (Fig 12).

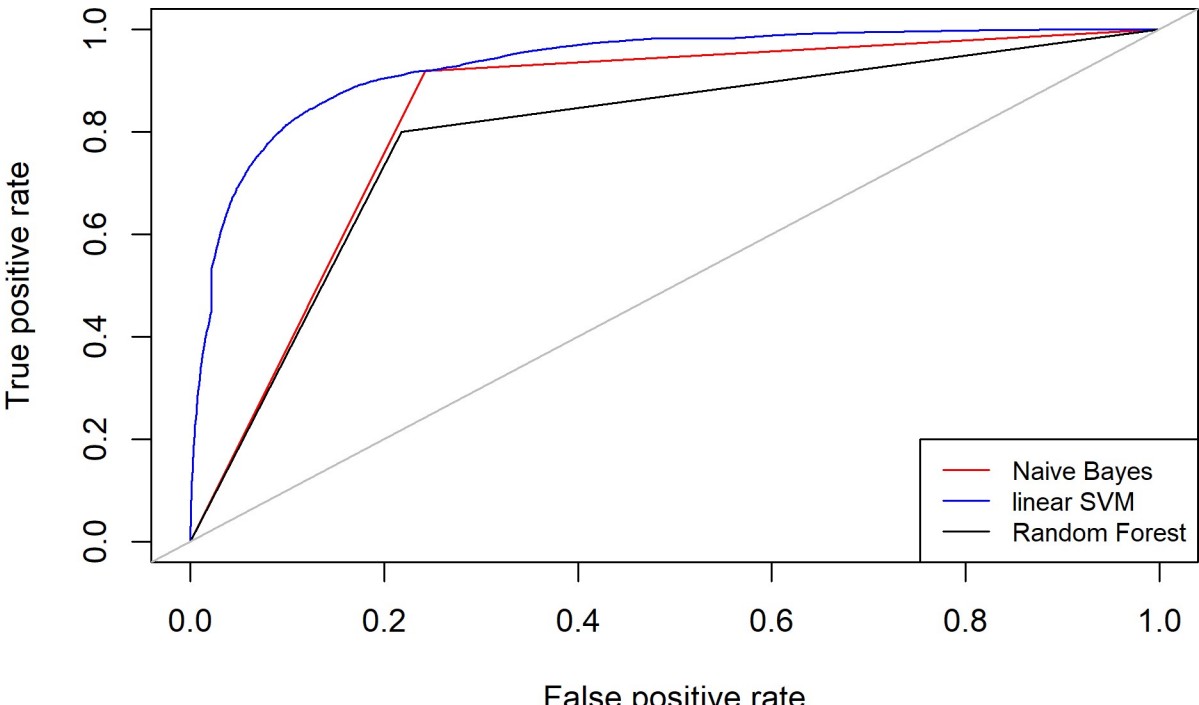

**Fig 12. ROC curves for accurately identifying members of the tech elite from the general Twitter population.**

In a nutshell, all these findings confirm that the tech elite communicates distinctively and is likely a "class for itself" with a distinct social identity. Our hypotheses mark particular features of this highly selected group. But even if these features do not identify the most relevant particularities of the tech elite, as we can only explain a fraction of the variance, we can easily identify them in the large pool of Twitter users.

## Discussion

The rapid dissemination of digital technologies catapulted the founders of large IT enterprises into the top ranks of wealth and power. Given their position of influence in contemporary and (probably) future societies, this paper explores the worldviews of the new tech elite. We focus on the richest 100 persons in tech identified by Forbes magazine. Elite research usually suffers from low to no response rates. Targeting a super-elite exacerbates the problem. In fact, we approached everyone from our list but got only one face-to-face interview. To circumnavigate this problematic access to data, we explored the digital traces of tech elite entrepreneurs to assess their distinctiveness relative to the general population and to other elites. To our knowledge, this is the first attempt to study tech elites with digitized text data. Social scientists have started to use social media data to scale ideologies of political elites, or citizens, but not of economic leaders [e.g., 84].

Specifically, we examine the worldviews of the IT tech elite with a simple "bag of words" approach. We hypothesized first that, as products of a society with strong meritocratic beliefs and frequently of elite institutions of higher education, the tech elite would see this world and future worlds in meritocratic terms. Our analysis of a large sample of their statements on Twitter (Tweets), relative to the general US Twitter-using population, indeed found that the tech elite tend to speak more frequently about merit-related topics and to more frequently use words that bespeak achievement-related concerns. They also speak more expansively and positively about the future than the general Twitter-using population. Our first hypothesis, proposing that the tech elite would see the world and the future in meritocratic, self-affirming, or even self-serving terms, was thus confirmed.

This paper provides a basis to directly test whether and how meritocratic beliefs translate into declining social mobility or even social closure in future research. A coherent, divisive and legitimizing social ideology is often seen as an important ingredient for class awareness and self-interested behavior [85]. "Career Funneling" at the most selective universities into tech jobs which are perceived as high status may provide a further explanation as to why tech clusters in Silicon Valley (re)produce a "class for itself" [e.g., 86].

Next, we hypothesized that the tech elite saw its endeavors in "entrepreneurial technoscience" as driven by a desire to "make the world a better place"—that is, that it is "mission-driven." We tested this hypothesis by comparing the philanthropic statements of members of the tech elite vis-a-vis the Giving Pledge and websites of their own foundations. In addition to comparing across elites, we compared age cohorts within the tech elite to see whether their ideas about "making the world a better place" and about philanthropy varied.

On this basis we found that the members of the tech elite were more likely than the other Pledgers to have an expansive and positive vision of philanthropic endeavor. The tech elite does, indeed, appear to have strong, positive sentiments toward the idea of "making the world a better place." Is this a "religious" inclination? Insofar as one can say based on these data, the answer to this question is "no." The tech elite in particular tended toward a more secular outlook. In addition, we found that the different age cohorts within the tech elite tended to stress different secular missions in regard to their philanthropic activities: the oldest, "hardware" generation emphasized "research"; the middle, "software" generation most often cited

"school"; and the youngest, "internet" generation most frequently used the word "can," a reflection perhaps of their youthful enthusiasm. In any case, educational and cultural missions aim to shape the public interest.

This leads to hypothesis 3, which examines the relationship of the tech elite to politics. It proposes that members of the elite have a contradictory relationship with democracy because market success and financial wealth should tend to entail worldviews and arguably activities (including philanthropic activities) that sidestep democratic representation. We found no statistically significant differences in whether or not the tech elite saw a positive relationship between power and money, or between power and democracy, as compared to the members of the US Twitter-using population. Yet, the tech elite denied that there is a positive connection between democracy and money, something that is logically inconsistent with the previous correlations and that is not shared by ordinary US Twitter users, who see the existence of a nexus between democracy and money.

Finally, we aim to determine directly whether the tech elite constitutes a "class for itself" in the sense that we can predict statistically whether a person is a member of the tech elite or not. Machine learning models indicate that we can do so quite accurately ($>$ 80%). Hence, the tech elite appears to be more than simply a part of the capitalist class in the broad sense of sharing "ownership of the means of production." Rather, members of the tech elite communicate similar worldviews and clearly form a distinct fraction of the capitalist class.

This study constitutes a first exploratory step in the analysis of the tech elite. We highlight three limitations that may inspire future research. First, we haven't been able in this research to trace everybody from our sample on Twitter and on foundation websites. Twitter is a competitor to other social media platforms like Facebook, Snapchat, Google Techies or WeChat. It is also blocked in China. Also, there is a digital divide between younger and older tech entrepreneurs. The majority of our "hardware" cohort does not use Twitter. For these reasons, the results may be less robust for older members of the tech elite and for Chinese members. Still, we can account for this selectivity and reach higher "response rates" than conventional random samples in elite studies [e.g., 45].

Secondly, we cannot rule out that the Twitter accounts are managed by professional PR experts. Still, we have replicated the analysis with Tweets from non-tech members of the Giving Pledge and see systematic differences between both privileged groups, who can equally afford professional support. Thus, the tech elite communicates differently, and even if the tech elite employs professional support these people are presumably articulating the views of their wealthy clients.

Finally, our insights into the democratic worldviews of the tech elite remain limited. We do not know if the tech elite's denial of a relationship between democracy and money is strategic communication or, in fact, their actual belief. Future research will have to explore this further. Social science research into social media use of political elites provides a promising roadmap for analyzing elite ideologies [e.g., 87, 88] and how they shape elite political behavior through donations and other exertions of financial power.

In conclusion, our research contributes to closing a research gap in societies with rising inequalities. We find that the 100 richest members of the tech world reveal distinctive attitudes that set them apart both from the general population and from other wealthy elites. As the companies they have created occupy a commanding position in the emerging tech-based economy, their views of our situation are likely to be of disproportionate significance. As a group, they are meritocratically inclined, concerned with the well-being of their fellow human beings, and relatively supportive of democratic society. Yet their position in a democratic system is contradictory: as a result of their enormous wealth, they have disproportionate influence over how discretionary income is spent. One need not be opposed to philanthropy to see that there

is a problem here. Future research will have to address whether the attitudes of this unusual group change over time, and whether policies can be found to bring their opportunities to shape social outcomes in line with a democratic social order.

## Supporting information

**S1 Table. Logistic regression on class.**
(DOCX)

## Author Contributions

**Conceptualization:** Hilke Brockmann, John Torpey.

**Data curation:** Hilke Brockmann, Wiebke Drews.

**Formal analysis:** Hilke Brockmann, Wiebke Drews.

**Investigation:** Hilke Brockmann, Wiebke Drews, John Torpey.

**Methodology:** Wiebke Drews.

**Project administration:** Hilke Brockmann.

**Resources:** Hilke Brockmann.

**Software:** Hilke Brockmann, Wiebke Drews.

**Supervision:** Hilke Brockmann, John Torpey.

**Validation:** Hilke Brockmann, Wiebke Drews.

**Visualization:** Hilke Brockmann, Wiebke Drews.

**Writing – original draft:** Hilke Brockmann, John Torpey.

**Writing – review & editing:** Hilke Brockmann, Wiebke Drews, John Torpey.

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
