## [Decision Letter · Decision Letter 0]

2 Jul 2020

PONE-D-20-07007

A Class for Itself? On the Worldviews of the New Tech Elite

PLOS ONE

Dear Dr. Brockmann,

Thank you for submitting your manuscript to PLOS ONE. After careful consideration, we feel that it has merit but does not fully meet PLOS ONE’s publication criteria as it currently stands. Therefore, we invite you to submit a revised version of the manuscript that addresses the points raised during the review process.

Please correct the manuscript according to the comments of all reviewers. Please answer all comments point by point.

We look forward to receiving your revised manuscript.

Kind regards,

Paweł Pławiak, D.Sc., Ph.D.

Academic Editor

PLOS ONE

Journal Requirements:

3. Please ensure that you refer to Figures 1, 9 and 10 in your text as, if accepted, production will need this reference to link the reader to the figure.

4. We note you have included a table to which you do not refer in the text of your manuscript. Please ensure that you refer to Tables 1 and 2 in your text; if accepted, production will need this reference to link the reader to the Tables.

Additional Editor Comments (if provided):

Please correct the manuscript according to the comments of all reviewers. Please answer all comments point by point.

Reviewers' comments:

Reviewer's Responses to Questions

**Comments to the Author**

1. Is the manuscript technically sound, and do the data support the conclusions?

Reviewer #1: No

Reviewer #2: Yes

2. Has the statistical analysis been performed appropriately and rigorously? 

Reviewer #1: No

Reviewer #2: Yes

3. Have the authors made all data underlying the findings in their manuscript fully available?

Reviewer #1: Yes

Reviewer #2: Yes

4. Is the manuscript presented in an intelligible fashion and written in standard English?

Reviewer #1: No

Reviewer #2: Yes

5. Review Comments to the Author

Reviewer #1: The common "bag-of-words" model, that ignores context and grammar nowdays is a simplest and not ehough comprehensive method for tasks described in the article. This mechanism is appropriate for text classification, indexing or searching, but not for semantic inference about meriocratic view of world.

Reviewer #2: The work applies sentiment analysis on Twitter data of tech elite. This work has useful results that can be investigated from a sociological perspective. In particular, as an application of using machine learning and NLP, I liked their work and their idea of sentiment classification (SC) on Twitter data for a special class of Twitter users. I think people can benefit from this study and this approach can be used in many other similar ways and led to many valuable results in sociological science.

In my opinion, this article may be published, but not in the form presented.

1) The related work excludes all the work done in sentiment analysis using machine learning approaches.

2) “Methods” section should be described better.

3) It is highly recommended that authors explain the classification algorithms used in the work more precisely.

4) Why SVM has been used for cross-validation and naïve Bayesian for classification?

5) In the “prediction” subsection of results, the authors talk about supervised model and supervised machine learning. How do the authors use these terms for two different purposes? To be exact, the authors say: “We use both a supervised model with our hypothesized topics (…) and supervised machine learning”. Later, the authors have said that the latter builds on a naïve Bayesian algorithm, while they have not mentioned the first one. Then in the last sentence of this paragraph they told that the naïve Bayesian model is more accurate. But what is the first algorithm that authors are comparing the naïve Bayesian model with? This part should be clarified in the text.

6) How do the authors compare their work with the previous studies in the field?

6. PLOS authors have the option to publish the peer review history of their article (what does this mean?). If published, this will include your full peer review and any attached files.

Reviewer #1: No

Reviewer #2: No

---

## [Author Response · Author response to Decision Letter 0]

26 Aug 2020

Thank you for the reviews and for your invitation to revise and resubmit our manuscript. Please find our answers to all comments and criticisms from the reviewers and the editor in our uploaded letter to the editor and the reviewers. The changes we made in the manuscript can be found in the attached revised version. 

I. We changed our manuscript according to the PLOS ONE style requirements. More specifically, we altered the headings, the labelling of the figures and tables, the formatting of the tables, the in-text citations, and the references.

II. Our data is available at (https://github.com/wiebkedrews/ClassForItself/releases/tag/v1.0)

III. We now refer to Figures 1, 9, and 10 in the text.

IV. We now refer to Table 1 and 2 in the text.

V. We uploaded a table in the Supporting Information files and update our in-text citations accordingly.

Our detailed responses to Reviewers #1 and #2 on questions 1, 2, 4, and 5 and be found in the letter to the reviewers and the editor. Both reviewers agreed that Question 3 had been satisfactorily addressed.

---

## [Decision Letter · Decision Letter 1]

22 Sep 2020

PONE-D-20-07007R1

A class for itself? On the worldviews of the new tech elite

PLOS ONE

Dear Dr. Brockmann,

Thank you for submitting your manuscript to PLOS ONE. After careful consideration, we feel that it has merit but does not fully meet PLOS ONE’s publication criteria as it currently stands. Therefore, we invite you to submit a revised version of the manuscript that addresses the points raised during the review process.

Please correct the manuscript according to all comments from reviewers. Please answer all comments from reviewers point by point.

We look forward to receiving your revised manuscript.

Kind regards,

Paweł Pławiak, D.Sc., Ph.D.

Academic Editor

PLOS ONE

Additional Editor Comments (if provided):

Please correct the manuscript according to all comments from reviewers. Please answer all comments from reviewers point by point.

Reviewers' comments:

Reviewer's Responses to Questions

**Comments to the Author**

1. If the authors have adequately addressed your comments raised in a previous round of review and you feel that this manuscript is now acceptable for publication, you may indicate that here to bypass the “Comments to the Author” section, enter your conflict of interest statement in the “Confidential to Editor” section, and submit your "Accept" recommendation.

Reviewer #1: (No Response)

Reviewer #2: All comments have been addressed

2. Is the manuscript technically sound, and do the data support the conclusions?

Reviewer #1: Yes

Reviewer #2: Yes

3. Has the statistical analysis been performed appropriately and rigorously? 

Reviewer #1: No

Reviewer #2: Yes

4. Have the authors made all data underlying the findings in their manuscript fully available?

Reviewer #1: Yes

Reviewer #2: Yes

5. Is the manuscript presented in an intelligible fashion and written in standard English?

Reviewer #1: Yes

Reviewer #2: Yes

6. Review Comments to the Author

Reviewer #1: There is in the article evaluation of the classification performed. However, there is no description about validation schema. It is strongly recomended to improve this section.

Reviewer #2: The authors have satisfactorily responded to all my concerns.

7. PLOS authors have the option to publish the peer review history of their article (what does this mean?). If published, this will include your full peer review and any attached files.

Reviewer #1: No

Reviewer #2: No

---

## [Author Response · Author response to Decision Letter 1]

9 Nov 2020

Dear Reviewers,

Thank you for the positive feedback. Reviewer 1 accepted the revised manuscript as it is. Reviewer 2 asks us to elaborate on our validation analysis. S/he criticizes in question 3 that our statistical analysis is not appropriately and rigorously preformed without further explications. However, in question 6, s/he answers: “There is in the article evaluation of the classification performed. However, there is not description about validation schema. It is strongly recomended (sic!) to improve this section.” We understand this critique as a request to elaborate on our classification analysis. We are happy to do this. We now explain our classification approach in more detail in the methods section, added further calculations, updated our code respectively and report additional evaluations and performance measures in the results section. More precisely, 

1. We subdivided the Methods section and added more information on our machine learning approach and its validation. The additions now read: 

“Machine Learning

 To determine if social media communication is fit and robust for elite research, we examine if we can distinguish the tech elite from the Twitter population. First and as a baseline, we run a logistic regression and regress class membership (tech elite vs US population) on the aforementioned content categories (future, merit, achieve, religion, democracy, power, money) as well as on positive and negative sentiments. Then we apply supervised machine learning techniques to classify Tweets in an automated process. Naïve Bayesian classifier (NB), Support Vector Machines (SVM), and Random Forests (RF) are commonly applied machine learning methods for text classification [see, e.g., 36]. We use NB, SVMs with linear kernel and with polynomial kernel, and fast implementation of RF for high dimensional data (R package ranger). (…)

We also test polynomial kernels. 

RF is an ensemble-based classifier that focuses on multiple decision trees to construct powerful prediction models. The method links bootstrap aggregation (bagging) with random feature selection to improve and decorrelate decision trees. After a random forest is generated, trees’ predictions are combined to classify new objects [50]. 

RF performs well on many problems, and can handle noisy and large data sets. However, they are not easy to interpret, which is why we use them here only as an additional validation measure. In comparison, NB performs particularly well with fewer training cases and short documents. SVM, provides better classification tasks with a large amount of training cases [63]. Our dataset consists of short but many texts. For these reasons, we run and compare different classifiers. 

Each classifier is trained on class labels (Tech elite, US population) with a randomly partitioned subset of Twitter data (training set, 80%), and predicts the most likely classes with the remaining unlabeled Tweets (test set, 20%). Moreover, to ensure the robustness of the SMV classifiers, each classification process has been repeated 10 times using different random splits into training and test data. A grid of parameter values was evaluated using cross-validation to obtain the optimally tuned SVM model. Finally, all classification results are evaluated by standard performance measures (accuracy, precision, recall), visualizations (ROC curve), and statistical tests (ANOVA). The code is published on Github.”

2. We elaborated further in the Results section on evaluation and performance measures: 

“Classification

In a final step, we test how well we can identify the class of the tech elite (vs non-tech elite). First, a logistic regression model reveals that all of the hypothesized topics (future, merit, achieve, religion, democracy, power, money, positive and negative emotions) are highly statistically significant in the determination of class membership (p<0.001). Nonetheless, all topics together predict only 5% of the overall variance (R²=.048) [Table in S1 Table]. We then test if supervised machine learning approaches with NB, SVM, and RF classifiers can predict if a person belongs to the tech elite or to the US population (for more details, see Methods section) without further content. For NB, we used a training set with 80% of the original Tweets. The confusion matrix in Table 2 displays the number of correctly and falsely classified Tweets. The overall accuracy is 83.8%, 91.8% of Tweets are predicted precisely (PPV not shown in the table) with a recall or sensitivity of 79.4%.

Table 2. Confusion Matrix and Performance Measures for Classified Twitter Data

 Naïve Bayesian

(N=100.000) Linear SVM (N=50.000) Random Forest

(N=100.000)

 Tech Population Tech Population Tech Population

Tech 9237 829 3710 670 7944 2176

Population 2391 7474 1278 4342 1988 7823

Accuracy 0.8384 0.8052 0.7911

Sensitivity 0.7944 0.7726 0.7998

Specificity 0.9002 0.8663 0.7824

AUC 0.8376 0.9313 0.7911

In comparison, the results for SVM with a linear kernel reveal similar numbers, even though we use only half of the cases for computational reasons. Overall, accuracy lies at 80.5% with a slightly lower PPV of 84.7% and a recall of 77.3%. 

Performance measures for a random forest classifier differ marginally. An ANOVA reveals no significant difference between the three classifier (F=1.24, p=0.35). Only an SVMs with polynomial kernel perform weakly why we excluded them from the further analysis. Our classifiers achieved an average area under the receiver operating characteristic curve (ROC) of 85.3 (AUC). 

Fig 11. ROC curves for accurately identifying members of the tech elite from the general Twitter population. 

In a nutshell, all these findings confirm that the tech elite communicates distinctively and is likely a “class for itself” with a distinct social identity.””

We believe that these changes have improved the paper substantially. Thank you again for your comments!

---

## [Decision Letter · Decision Letter 2]

3 Dec 2020

A class for itself? On the worldviews of the new tech elite

PONE-D-20-07007R2

Dear Dr. Brockmann,

We’re pleased to inform you that your manuscript has been judged scientifically suitable for publication and will be formally accepted for publication once it meets all outstanding technical requirements.

Kind regards,

Paweł Pławiak, D.Sc., Ph.D.

Academic Editor

PLOS ONE

Additional Editor Comments (optional):

Reviewers' comments:

Reviewer's Responses to Questions

**Comments to the Author**

1. If the authors have adequately addressed your comments raised in a previous round of review and you feel that this manuscript is now acceptable for publication, you may indicate that here to bypass the “Comments to the Author” section, enter your conflict of interest statement in the “Confidential to Editor” section, and submit your "Accept" recommendation.

Reviewer #2: All comments have been addressed

2. Is the manuscript technically sound, and do the data support the conclusions?

Reviewer #2: (No Response)

3. Has the statistical analysis been performed appropriately and rigorously? 

Reviewer #2: Yes

4. Have the authors made all data underlying the findings in their manuscript fully available?

Reviewer #2: Yes

5. Is the manuscript presented in an intelligible fashion and written in standard English?

Reviewer #2: Yes

6. Review Comments to the Author

Reviewer #2: The authors have addressed all the concerns. Particularly, the method and comparison sections have been improved and the paper has appropriate information and interesting results.

7. PLOS authors have the option to publish the peer review history of their article (what does this mean?). If published, this will include your full peer review and any attached files.

Reviewer #2: No

---

## [Editor Report · Acceptance letter]

7 Dec 2020

PONE-D-20-07007R2 

A class for itself?On the worldviews of the new tech elite 

Dear Dr. Brockmann:

I'm pleased to inform you that your manuscript has been deemed suitable for publication in PLOS ONE. Congratulations! Your manuscript is now with our production department. 

Kind regards, 

on behalf of

Prof. Paweł Pławiak 

Academic Editor

PLOS ONE